# Can Intraoperative Intra-Articular Loads Predict Postoperative Knee Joint Laxity Following Total Knee Arthroplasty? A Cadaver Study with Smart Tibial Trays

**DOI:** 10.3390/s21155078

**Published:** 2021-07-27

**Authors:** Darshan S. Shah, Orçun Taylan, Matthias Verstraete, Pieter Berger, Hilde Vandenneucker, Lennart Scheys

**Affiliations:** 1Department of Development and Regeneration, Institute for Orthopaedic Research and Training (IORT), KU Leuven, 3000 Leuven, Belgium; orcun.taylan@keuleuven.be (O.T.); hilde.vandenneucker@uzleuven.be (H.V.); lennart.scheys@kuleuven.be (L.S.); 2Department of Mechanical Engineering, Indian Institute of Technology Bombay, Mumbai 400076, India; 3Stryker European Operations Ltd., 1101 CM Amsterdam, The Netherlands; matthias.verstraete@stryker.com; 4Department of Orthopaedics, University Hospital Leuven, 3000 Leuven, Belgium; pieter.berger@uzleuven.be

**Keywords:** joint balancing, physiological simulator, instrumented insert, in vitro testing, squatting, active vs. passive

## Abstract

Ligament balancing during total knee arthroplasty (TKA) often relies on subjective surgeon experience. Although instrumented tibial trays facilitate an objective assessment of intraoperative joint balance through quantification of intra-articular joint loads, postoperative clinical assessment of joint balance relies on passive stress tests quantifying varus–valgus joint laxity. This study aimed at correlating the intraoperative and postoperative metrics used to assess joint balance while also comparing joint loads obtained during passive assessment and active functional motions. Four experienced surgical fellows were assigned a fresh-frozen lower limb each to plan and perform posterior-stabilised TKA. An instrumented tibial insert measured intraoperative intra-articular loads. Specimens were then subjected to passive flexion–extension, open-chain extension, active squatting, and varus–valgus laxity tests on a validated knee simulator. Intra-articular loads were recorded using the instrumented insert and tibiofemoral kinematics using an optical motion capture system. A negative correlation was observed between mean intraoperative intra-articular loads and corresponding mean postoperative tibial abduction angle during laxity tests (medial: R = −0.93, *p* = 0.02; lateral: R = −0.88, *p* = 0.04); however, this was not observed for each specimen. Peak intra-articular load distribution for active squatting was lateral-heavy, contrasting to the medial-heavy distribution observed in passive intraoperative measurements, for all specimens. These aspects should be given due consideration while assessing intraoperative and postoperative joint stability following TKA.

## 1. Introduction

Total knee arthroplasty (TKA) is an established surgical procedure to treat severe osteoarthritis in the knee; however, joint instability has been reported as a major cause of postoperative patient dissatisfaction [1]. Soft-tissue balancing, although vital to achieve joint stability during TKA, often relies on subjective surgeon feel and experience intraoperatively [2]. Advancements in surgical instrumentation and navigation have led to objective assessment of soft-tissue balance using spacer blocks and manual distraction devices indicating joint gaps at fixed flexion angles [3,4]. Electronic sensor-based instrumented tibial components have improved this objective assessment by facilitating the direct measurement of intra-articular loads over a continuous range of knee flexion [5,6].

Verasense (OrthoSensor, Dania Beach, FL, USA), a smart tibial tray compatible with already existing TKA designs, facilitates the quantitative measurement of soft-tissue balancing during TKA by measuring intraoperative loads in situ in the medial and lateral compartments at the points of contact between the femoral condyles and tibial plateaux [6,7]. With real-time feedback of the rotational alignment of the implant components, this instrumented insert has been shown to improve postoperative patient scores and patient satisfaction, thereby strengthening the potential importance of achieving optimal joint balance during TKA [7].

While intra-articular loads are used to achieve joint stability intraoperatively, clinical assessment of postoperative joint stability typically relies on passive tests, such as the stress tests quantifying the varus–valgus laxity of the knee at multiple flexion angles [8], performed manually and subjectively by the surgeon. Thus, although two distinct parameters—intraoperative intra-articular loads and postoperative joint laxity—are used to achieve a common goal of improving joint stability, they have not been correlated in the literature.

Moreover, while this manual passive postoperative assessment of joint stability is performed at constant flexion angles, the aim is to have desired postoperative joint stability for dynamic activities of daily living, such as walking, over the entire range of knee flexion. Active motions differ from their passive counterparts primarily owing to active muscle force production, as well as their often closed-chain nature resulting in higher joint contact forces [9].

Therefore, the primary aim of this in vitro study was to correlate intraoperative intra-articular loads to the postoperative tibiofemoral abduction measured during the varus–valgus stress tests, hypothesizing a negative correlation between the two parameters. The secondary aim of this study was to compare intra-articular loads measured intraoperatively to those measured postoperatively during passive and active motions, with the hypothesis that active motions would alter load distribution between the medial and lateral compartments owing to a change in joint dynamics and muscle force production.

## 2. Materials and Methods

### 2.1. Specimen Preparation

Four fresh-frozen cadaveric specimens (two bilateral lower limbs: male, age = 75 years, height = 1.78 m, bodyweight = 83 kg, and male, age = 91 years, height = 1.76 m, bodyweight = 78 kg) were obtained from the institute body donation programme following ethical approval by the local ethics committee. None of the specimens had signs of lower limb disorder or prior surgical intervention.

Magnetic resonance imaging and full leg radiographs were obtained for each specimen to design specimen-specific cutting blocks for the TKA surgery using the VISIONAIRE protocol (VISIONAIRE II, Smith & Nephew, Memphis, TN, USA). Rigid marker frames with reflective spheres (diameter = 12 mm) were attached to the femur and tibia using bicortical bone pins. Computed-tomography (CT) scans (slice thickness = 0.6 mm; Siemens SOMATOM Force, Siemens Healthineers, Erlangen, Germany) obtained for each frozen specimen in full extension were used to identify the location of the markers relative to anatomic landmarks (Mimics 20.0, Materialise, Leuven, Belgium) in order to define a joint coordinate system for the femur and tibia [10].

Each lower limb was thawed for 24 h before resecting it 32 cm proximally and 28 cm distally to the knee joint. Care was taken to preserve the joint capsule, ligaments, and tendons while dissecting the surrounding skin and subcutaneous tissue. The femur and tibia were embedded in metal pots using acrylic resin (Struers, Ballerup, Denmark) in a physiologic orientation. Tendons of the quadriceps and the hamstrings were carefully exposed and prepared to be attached to an electromechanical actuator for dynamic control and passive springs for constant tensile load application, respectively.

### 2.2. TKA Surgery

Each of the four specimens was randomly allotted to one of four experienced surgical fellows (Objective Structured Assessment of Technical Skills (OSATS) scores > 85%) [11,12]. Surgeons performed posterior-stabilised TKA (Legion, Smith & Nephew, Memphis, TN, USA) on their corresponding specimens based on the preoperative surgical plan (VISIONAIRE II, Smith & Nephew, Memphis, TN, USA) designed by them for their respective specimens. Following optimum implant sizing, component alignment, and ligament release, if needed, to attain balance as per standard norms [2,3], the surgeons used an instrumented tibial insert (Verasense, OrthoSensor, Dania Beach, FL, USA) to assess the achieved joint balance in the coronal and sagittal planes at fixed flexion angles (0°, 10°, 30°, 45°, 60°, 90°). Implant sizing and alignment were not altered regardless of the outcome of the joint balance assessment.

### 2.3. Experimental Testing

Following implantation, each specimen was mounted on a previously validated physiological knee-joint simulator [13] and subjected to passive flexion–extension, open-chain extension, active squatting, and varus–valgus stress tests. Passive flexion–extension involved manual movement of the unconstrained tibia with respect to the affixed femur from complete extension (0°) to deep flexion (160°). Open-chain extension was performed using an electromechanical actuator connected to the quadriceps tendon to actively extend the knee from mid-flexion (90°) to complete extension (0°), while a 3 kg dead weight was attached to the distal tibia to replicate the weight of the resected foot. Active squatting involved imposing a vertical displacement on the proximal femur to achieve a cyclic knee motion (35–100°), while applying a dynamic physiological quadriceps load using the electromechanical actuator to generate and maintain a vertical ankle force of 110 N and attaching the medial and lateral hamstrings to 50 N constant force springs. Varus–valgus stress tests were used to assess specimen laxity at discrete flexion angles (0°, 30°, 60°, 90°) by subjecting joint to an adduction–abduction moment of 10 N.m. This was achieved by affixing the femur rigidly to the simulator assembly and using a handheld force gauge (resolution = 0.1 N; Series 4, Mark-10, Copiague, NY, USA) to manually apply a tensile force at the distal end of the unconstrained tibia perpendicular to the tibial long axis.

The specimen was kept moist during experimental testing with phosphate-buffered saline solution to mitigate tissue-drying effects. The order of tasks performed was randomized to avoid potential bias. Each task was performed in triplicate by a single operator.

A six-camera motion capture system (capture frequency = 100 Hz, MX40 cameras with standard passive calibration wand; Vicon, Oxford, UK) was used to track the motion of the femur and tibia. Intra-articular loads in the medial and lateral compartments of the knee were recorded during all tasks using the custom company software (Verasense, OrthoSensor, Dania Beach, FL, USA).

### 2.4. Data Analysis

The recorded trajectories of markers on the femur and tibia were processed further (Nexus 2.9, Vicon, Oxford, UK) to calculate tibiofemoral kinematics for each specimen using a custom code (MATLAB R2018b, MathWorks Inc, Natick, MA, USA) based on joint coordinate systems defined using anatomical landmarks obtained from CT. The relationship between postoperative joint laxity, measured using the tibial abduction angle during varus–valgus stress tests, and intraoperative intra-articular load was assessed for each specimen using the Pearson correlation test (*p* < 0.05). Peak intra-articular loads in the medial and lateral compartments were calculated for passive flexion–extension, open-chain extension, and active squatting, and compared to values obtained intraoperatively.

## 3. Results

### 3.1. Intraoperative Intra-Articular Loads

Intra-articular loads measured intraoperatively decreased with increasing knee flexion angle in both the medial and lateral compartments (Figure 1). Medial and lateral intra-articular loads reduced in all specimens by more than 72% and 95%, respectively, at 90° knee flexion, as compared to values measured in complete extension. The lateral load was found to be lower than the medial load throughout the range of flexion for three out of the four specimens.

### 3.2. Postoperative Laxity vs. Intraoperative Intra-Articular Load

Varus–valgus stress tests revealed greater stiffness in extension (1.7 ± 1.2°) than in flexion (4.4 ± 2.4°) postoperatively (Figure 2). Mean postoperative tibial adduction and abduction across specimens exhibited a negative correlation to intraoperative intra-articular loads in the medial and lateral compartments, respectively (R_med_ = −0.93, *p* = 0.02; R_lat_ = −0.88, *p* = 0.04) (Table 1).

### 3.3. Peak Intra-Articular Load

Peak total intra-articular load across the range of knee flexion was higher for actively loaded tasks—open-chain extension and squatting—as compared to passive measurements, with the peak load during squatting being the highest amongst all tasks for three out of the four specimens (Figure 3). For all specimens, peak loads were observed in extension when measured intraoperatively (0°) and during passive flexion–extension (0°) and open-chain extension (10°); however, peak loads during active squatting were observed in deep flexion (100°).

In the case of peak loads for intraoperative measurements, passive flexion–extension, and open-chain extension, loads in the medial compartment were higher than those in the lateral compartment for three out of the four specimens; however, for active squatting, loads in the lateral compartment were higher than those in the medial compartment for all specimens (Figure 4).

## 4. Discussion

Ligament balancing is vital in the successful outcome of TKA [2]; however, parameters to assess joint balance intraoperatively are starkly different from those used during postoperative assessment. Moreover, postoperative joint stability is largely assessed manually at static flexion angles which contrasts the goal of achieving desired joint stability during loaded functional activities [8]. This cadaveric study not only measured and correlated intraoperative intra-articular load—a reliable metric to assess joint balance during TKA [6]—to postoperative joint laxity but also compared intraoperative loads measured at static flexion angles to postoperative loads during dynamic functional tasks using a physiological simulator.

One of the highlights of this study was the participation of four experienced surgical fellows, each performing specimen-specific TKA. The same implant was used by all fellows to avoid implant-specific variability in postoperative biomechanics; however, specimen-specific cutting guides were based on the preoperative surgical planning performed by each fellow. Moreover, although trained at different institutions globally, the fellows followed standard surgical norms of joint balancing during TKA [2,3]. This induced the all-important surgeon-specific inter-subject variability.

Intra-articular loads measured during TKA revealed that all specimens were tighter in extension, thereby leading to higher loads, than in flexion (Figure 1). Moreover, the medial compartment was tighter than the lateral compartment for three out of the four specimens. These observations contrasted the standard norms followed by the surgical fellows, wherein they aimed to achieve a symmetric balance between the medial and lateral compartments in both extension and flexion [2,3]. This contrast in the imbalance of intra-articular loads, despite the subjective adherence to standard norms for balancing made by each fellow, elucidates the importance of objectifying intraoperative joint balancing.

Postoperative joint laxity largely corroborated with intraoperative intra-articular loads, since it was ensured that implant sizes and/or component alignments were not altered following intraoperative joint balancing assessment with the instrumented tibial trays. Postoperative tibial adduction and abduction during the varus and valgus stress tests, respectively, were higher in flexion than in extension (Figure 2), thereby indicating specimens being tighter in extension than flexion. Mean postoperative joint laxity during the varus and valgus stress tests had a significant negative correlation to mean intraoperative intra-articular loads in the medial and lateral compartments, respectively (Table 1); however, contrary to our hypothesis, this was not the case for each individual specimen. Despite a decrease in intra-articular loads being associated with an increase in varus–valgus laxity with increasing knee flexion, a significant correlation was observed for only one out of the four specimens in both the medial (varus) and lateral (valgus) aspects (Table 1). One of the primary reasons for this discrepancy might lie in the definition of the joint balancing metric used intraoperatively and postoperatively—while the former relies on loads in condylar compartments without the joint being subjected to an external load, the latter is based on joint kinematics following the application of an external moment on the joint replicating postoperative clinical assessment [14,15]. Another reason for a weak statistical correlation between the two parameters, despite a strong subjective conformance associated over the range of flexion, could be the dearth of static flexion angles used to test postoperative joint laxity; a larger dataset consisting of interpolating flexion angles will be sought for in future studies to improve the evaluation of the aforementioned correlation.

Peak total intra-articular load over the range of knee flexion was higher for dynamic loaded tasks as compared to static measurements (Figure 3), with the peak total load being highest for squatting in most cases, as hypothesized. More importantly, the peak load during squatting was observed in deep flexion, as opposed to extension during all other tasks, including the intraoperative measurements at static knee flexion angles. Furthermore, the distribution of intra-articular loads between the medial and lateral compartments changed drastically in the case of squatting—from a medial-heavy to a lateral-heavy load distribution for most specimens (Figure 4). The higher magnitude of loads, their occurrence at higher flexion angles, and the altered load distributions encountered during squatting could be attributed to the closed-chain nature of the motion, wherein the distal reaction forces contribute to additional intra-articular joint loads, coupled with active co-contraction of the quadriceps and hamstrings [13].

However, this was not the case for one specimen (Spec 2), which exhibited such high intra-articular loads during intraoperative measurements already that the differences in peak loads between passive and active motions were minimal, as compared to those observed in other specimens (Figure 3). Moreover, the lateral-heavy load distribution for this specimen was similar across passive and active tasks but starkly contrasting to all other specimens, which shifted from a medial-heavy to lateral-heavy distribution only during squatting (Figure 4). A preliminary inference drawn from this finding could be that a tight joint might be able to perform passive and active tasks in a repeatable manner; although, the intra-articular loads associated with this level of tightness are not clinically recommended [16,17]. The differences in intra-articular load distribution for passive and active motions, and the potential trade-off of a tight joint, should be considered while performing a joint balancing procedure during TKA, as well as clinically assessing postoperative joint stability.

Although this unique cadaveric simulator study included state-of-the-art instrumented sensors, one of the main shortcomings of the study was the small sample size. Future studies should not only look to include a larger number of specimens but also increase the number of static flexion angles used to test intraoperative intra-articular loads and corresponding postoperative joint laxity, thereby facilitating a better correlation analysis of the two parameters with statistical and clinical significance. Another limitation was the underlying assumption that Verasense sensors could sustain intra-articular pressure encountered during the actively loaded squatting motion, often reaching ~16 MPa [18], without damage. While no physical or electronic damage was reported to any sensor for the entire duration of the surgery and the experimental testing, it must be noted that the instrumented tibial inserts were commercially calibrated only until 310 N in each compartment, while loads reported, especially for the active motions, were much higher (Figure 4). Further developmental research is required to improve the ratings of the smart tibial sensors to facilitate the assessment of joint contact forces during functional motions.

## 5. Conclusions

A conformance between intraoperative intra-articular joint loads and postoperative joint laxity was observed, although the correlation between these parameters was not statistically significant for all specimens. Moreover, there was a stark discrepancy in peak joint load distribution across the medial and lateral compartments between passive tasks and active functional motions. These aspects should be given due consideration while performing a joint balancing procedure during TKA, as well as clinically assessing postoperative joint stability.

## Figures and Tables

**Figure 1 sensors-21-05078-f001:**
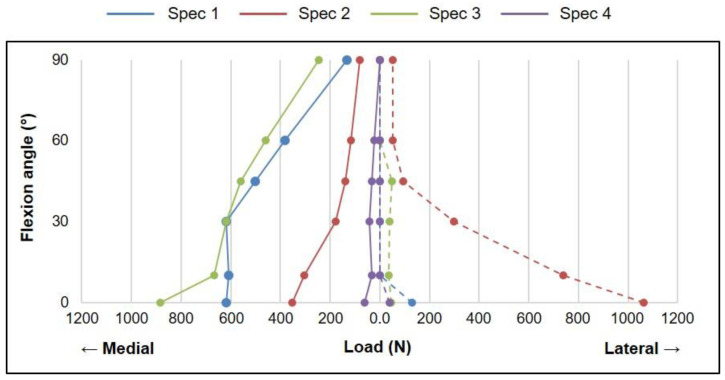
Intraoperative intra-articular loads in the medial (solid) and lateral (dashed) compartments of each specimen reduced with increasing knee flexion.

**Figure 2 sensors-21-05078-f002:**
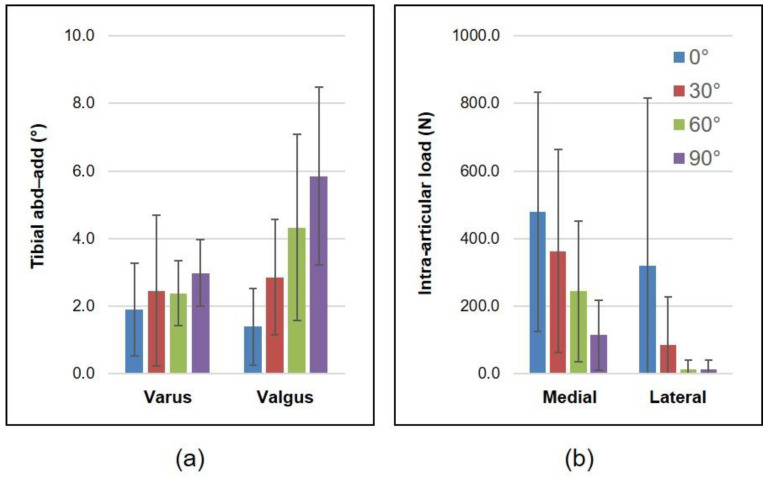
(**a**) Tibial abduction–adduction during the varus and valgus laxity tests measured postoperatively and (**b**) intra-articular loads in the medial and lateral compartments measured intraoperatively for increasing knee flexion angles (data represented as mean ± standard deviation across four specimens).

**Figure 3 sensors-21-05078-f003:**
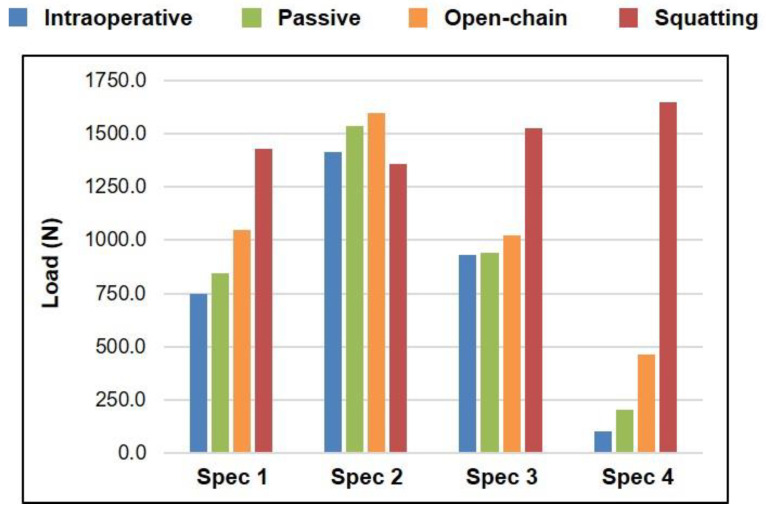
Peak total intra-articular load (sum of loads measured in the medial and lateral compartments) across the range of knee flexion for each specimen measured intraoperatively (blue), and during passive flexion–extension (green), open-chain extension (orange), and active squatting (red). Peak load was observed in extension when measured intraoperatively (0°) and during passive flexion–extension (0°) and open-chain extension (10°), while peak load during squatting was observed in deep flexion (100°).

**Figure 4 sensors-21-05078-f004:**
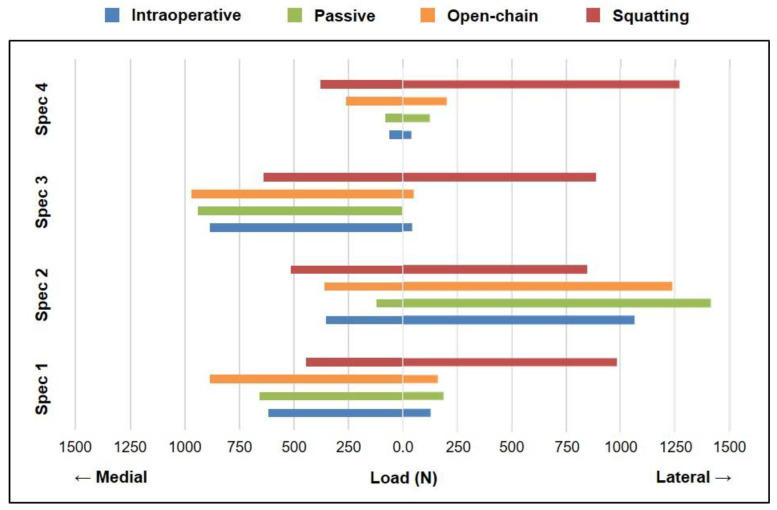
Peak total intra-articular load distribution in the medial (left) and lateral (right) compartments across the range of knee flexion for each specimen measured intraoperatively (blue) and during passive flexion–extension (green), open-chain extension (orange), and active squatting (red). Peak load was observed in extension when measured intraoperatively (0°) and during passive flexion–extension (0°) and open-chain extension (10°), while peak load during squatting was observed in deep flexion (100°).

**Table 1 sensors-21-05078-t001:** Correlating the postoperative tibial abduction measured during the varus (Var) and valgus (Val) stress tests with the intraoperative intra-articular loads in the medial (Med) and lateral (Lat) compartments, respectively, at fixed knee flexion angles (Flex) for each specimen and consolidated for the means across all specimen (Pearson’s correlation coefficient (R); statistical significance: *p* < 0.05).

		Spec 1		Spec 2		Spec 3		Spec 4		Mean (SD)	
**Postoperative laxity**	**Flex**	**Var**	**Val**	**Var**	**Val**	**Var**	**Val**	**Var**	**Val**	**Var**	**Val**
	0°	1.2	0.9	0.5	0.6	2.2	0.9	3.7	3.1	1.9 (1.4)	1.4 (1.1)
	30°	1.4	1.7	0.6	1.1	2.2	4.0	5.6	4.6	2.5 (2.2)	2.9 (1.7)
	60°	1.8	1.9	2.0	2.1	1.9	5.9	3.8	7.4	2.4 (1.0)	4.3 (2.7)
	90°	2.3	2.5	3.0	6.2	2.3	5.9	4.4	8.9	3.0 (1.0)	5.9 (2.6)
**Intraoperative loads**	**Flex**	**Med**	**Lat**	**Med**	**Lat**	**Med**	**Lat**	**Med**	**Lat**	**Med**	**Lat**
	0°	618.3	129.0	351.4	1063.1	885.2	44.5	62.3	40.0	479.3 (353.2)	319.1 (497.6)
	30°	618.3	0.0	177.9	298.0	618.3	40.0	40.0	0.0	363.6 (299.4)	84.5 (143.6)
	60°	382.5	0.0	115.6	53.4	458.1	0.0	22.2	0.0	244.6 (208.7)	13.3 (26.7)
	90°	133.4	0.0	80.1	53.4	244.6	0.0	0.0	0.0	114.5 (102.6)	13.3 (26.7)
**R**		−0.99	−0.86	−0.80	−0.62	−0.08	−0.88	−0.07	−0.74	−0.93	−0.88
***p*** **-value**		<0.01	0.05	0.08	0.17	0.35	0.04	0.35	0.11	0.02	0.04

## Data Availability

The data presented in this study are available on request from the corresponding author. The data are not publicly available due to ethical and privacy considerations associated with human cadaveric donor material.

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
