# Peer review of "Can Intraoperative Intra-Articular Loads Predict Postoperative Knee Joint Laxity Following Total Knee Arthroplasty? A Cadaver Study with Smart Tibial Trays"

_sensors, 2021, doi:10.3390/s21155078_

Round 1

Reviewer 1 Report

Dear authors,

I would like to congratulate you for this excellent manuscript. The topic and the main reflection about the importance of the relationship between gap balancing and probable clinical loads, will be really useful for knee surgeons.

Even I like to remark all points that I don't agree, it is difficult to find some errors in structure and methodology in the present article. So, mainly, I will try to make some reflections about the limitations I think you have to consider in order to add in the discussion.

It is probably a poor study in terms of sample size. I suppose, authors want to show the negative correlation 'tendency' between intra-operative loads and post-operative laxity. However, if you want to see a better and consistent results, I propose to the authors to increase the sample size.

The second weak point, that I can reflect about, is the subjectivity in the evaluation of the results, taking into account that each one of the four specimens has been operated for a different surgeon. Of course they were trained to perform the same procedure, but all of them have a different way to understand ligament balance in terms of feeling intra-operatively. The results showed in Table 1, let us know that each surgeon goes for a different final laxity. This situation also could be because of the thickness of the tibial insert that the surgeon wants to implant. However, I understand that the aim of the present study is to analyse a correlation, despite of the individual way to balance the knee. In this point, my reflection is that you need more specimens to purpose us a solid conclusion.

There are some differences between the extension and flexion gaps. Three of the four specimens have a clear imbalance in the flexion gap, so I am not really sure about the correlation you explain. so, you have to talk about laxity and loads, but not about extension and flexion. I mean, if you want to talk about the correlation in different degrees of flexion you have to achieve the same equilibrium in those degrees of movement, because it is obvious that when we have more laxity loads goes down. Nevertheless, laxity could be a problem in the future survival of the implant, not for the increased loads, but for the increased abnormal friction with the normal gait cycles. 

Finally I would like to ask you for the primitive laxity and shape/axis of the four specimens. Probably could be interesting to correlate those results with the native pre-operative deformity. 

Reviewer 2 Report

Thank you for the opportunity to review this manuscript, which considers some interesting, applied issues. This study appears to be novel, but as submitted needs considerable work on the presentation. The authors showed an interesting point about the “"Can intraoperative intra-articular loads predict postoperative 2 knee joint laxity following total knee arthroplasty?”, unfortunately, there are several hard points to overcome.

Abstract

The authors need to include the significative data relate to the aim of this study

Method section

The authors should clarify the data “75yrs, 1.78m, 83kg” which variables?

Please to justify how the flexion angles was calculated/analyzed

About the “A six-camera motion capture system” the authors need to clarify the calibration and markers used

Unclear the means of Spec 1,2,3,4 !! please to justify in the main document

Please to add the limitations; for example the sample size/power/reliability of the measures
